# Smell and taste changes are early indicators of the COVID-19 pandemic and political decision effectiveness

Denis Pierron [1,13,14✉], Veronica Pereda-Loth [1,13,14], Marylou Mantel [2], Maëlle Moranges [2], Emmanuelle Bignon[3], Omar Alva[1], Julie Kabous[1], Margit Heiske[1], Jody Pacalon[3], Renaud David[4], Caterina Dinnella[5], Sara Spinelli[5], Erminio Monteleone[5], Michael C. Farruggia [6], Keiland W. Cooper [7], Elizabeth A. Sell [8], Thierry Thomas-Danguin [9], Alyssa J. Bakke [10], Valentina Parma [11], John E. Hayes[10], Thierry Letellier[1], Camille Ferdenzi [2,13,14], Jérôme Golebiowski [3,12,13,14✉] & Moustafa Bensafi [2,13,14✉]

In response to the COVID-19 pandemic, many governments have taken drastic measures to avoid an overflow of intensive care units. Accurate metrics of disease spread are critical for the reopening strategies. Here, we show that self-reports of smell/taste changes are more closely associated with hospital overload and are earlier markers of the spread of infection of SARS-CoV-2 than current governmental indicators. We also report a decrease in self-reports of new onset smell/taste changes as early as 5 days after lockdown enforcement. Cross-country comparisons demonstrate that countries that adopted the most stringent lockdown measures had faster declines in new reports of smell/taste changes following lockdown than a country that adopted less stringent lockdown measures. We propose that an increase in the incidence of sudden smell and taste change in the general population may be used as an indicator of COVID-19 spread in the population.

[1] Équipe de Médecine Evolutive Faculté de chirurgie dentaire; UMR5288; CNRS/Université Paul-Sabiater Toulouse III, Toulouse 31400, France. [2] Lyon Neuroscience Research Center, CNRS UMR5292, INSERM U1028, University Claude Bernard Lyon 1, Bron, France. [3] Université Côte d'Azur, CNRS, Institut de Chimie de Nice UMR7272, Nice, France. [4] Université Côte d'Azur, CHU de Nice, Nice Memory Clinic, Nice, France. [5] University of Florence, Florence, Italy. [6] Interdepartmental Neuroscience Program, Yale University, 333 Cedar Street, New Haven, CT 06520, USA. [7] Department of Neurobiology and Behavior, University of California, Irvine, CA 92697, USA. [8] Perelman School of Medicine, University of Pennsylvania, 3400 Civic Center Blvd, Philadelphia, PA 19104, USA. [9] Centre des Sciences du Goût et de l'Alimentation, INRAE, CNRS, AgroSup-Dijon, University Bourgogne Franche-Comté, Dijon, France. [10] The Pennsylvania State University, Philadelphia, PA 19104, USA. [11] Temple University, Philadelphia, PA 19122, USA. [12] Department of Brain and Cognitive Sciences, Daegu Gyeongbuk Institute of Science and Technology, Daegu 711-873, South Korea. [13] These authors contributed equally: Denis Pierron, Veronica Pereda-Loth, Camille Ferdenzi, Jérôme Golebiowski, Moustafa Bensafi. [14] These authors jointly supervised this work: Denis Pierron, Veronica Pereda-Loth, Camille Ferdenzi, Jérôme Golebiowski, Moustafa Bensafi. ✉email: denis.pierron@univ-tlse3.fr; jerome.golebiowski@univ-cotedazur.fr; moustafa.bensafi@cnrs.fr

Following similar decisions in China and Italy, a strict lockdown was enforced in France beginning on March 17, 2020 to block the progression of COVID-19 and alleviate pressure on hospitals. One issue currently faced by governments is how to conduct the progressive relaxation of the lockdown[1], which needs to be conducted systematically and carefully to prevent subsequent outbreaks while facilitating economic activity and recovery. On May 7, 2020, the French government categorized each geographical area as being red or green, depending on their COVID-19 prevalence. Compared to green areas, red areas were characterized by: (i) higher active circulation of the virus, (ii) higher level of pressure on hospitals (i.e., CCRU occupancy), and (iii) reduced capacity to test new cases (Fig. 1a). In each area, red/green labels were used to define steps associated with the local relaxation of lockdown. The French Ministry of Health used the ratio of consultations for suspected cases of COVID-19 to general consultations at the emergency room (ER) in hospitals as an indicator to assess the active circulation of the virus (detailed in "Methods" section). Concurrently, changes in smell and taste are prominent symptoms of COVID-19[2–5], as has consistently. been demonstrated in many countries (e.g., Iran[6], Spain[7], France[8], Italy[9], Germany[10], and the UK[2], among others). More critically, these chemosensory changes generally occur earlier than other symptoms[9] and may constitute more specific symptoms than fever or dry cough[2,11]. Accordingly, monitoring self-reported changes in smell and taste could thus provide early and specific information on the spread of COVID-19 in the general population and support health system monitoring to avoid daily CCRU admission overflows. Using data from a global, crowd-sourced study deployed in 30+ languages (Global Consortium for Chemosensory Research survey, GCCR, see "Methods" section), we tested whether changes in smell/taste at the population level could be used as an early indicator for local COVID-19 outbreaks. As pre-registered (see "Methods" section), our primary aim was to test the association between self-reported smell and taste changes and indicators of pressure in hospitals (COVID-related hospitalizations, CCRU admissions, and mortality rates) for each French administrative region over the last 3 months. Our secondary aim was to examine temporal relationships between the peak of smell and taste changes in the population and the peak of COVID-19 cases and the application of lockdown measures. The potential for self-reported smell and taste loss to serve as an early indicator of the number of COVID-19 cases—and hence hospital stress—was tested in a natural experiment by comparing France with Italy and the UK, which implemented lockdown with different timing and levels of stringency. Here, we show that self-reports of smell/taste changes are closely associated with hospital overload and are early markers of the spread of infection of SARS-CoV-2.

## Results

### Changes in smell and taste are associated with overwhelmed healthcare systems.
The relationship between self-reported changes in smell and taste by French residents (diagnosed as COVID-19+ or not, see "Methods" section and Supplementary Table 1) and estimators of local healthcare system stress was evaluated geographically. Figure 1a depicts the geographical distribution in red and green regions (as defined by the French government) and participants who self-reported changes in their smell and taste. Red areas of France account for 40.8% of the population. Green areas are clustered into a group with both a low number of self-reported chemosensory changes and a low number of admissions to CCRUs (Fig. 1b). Red areas show an opposite trend (Chi-square $<1 \times 10^{-200}$ and Biserial correlations $p < 1.3 \times 10^{-2}$). A strong relationship exists between self-reported changes in smell and taste and the number of admissions to CCRUs ($R_{smell} = 0.88$, $p = 8.9 \times 10^{-08}$). This correlation remained significant even after removing the two most impacted areas (Alsace and Ile de France, $R_{smell} = 0.72$; $p < 3 \times 10^{-04}$), indicating that the significant relationship is not driven solely by these two regions.

Strikingly, use of self-reported chemosensory changes produced a stronger correlation than the current governmental indicator of virus circulation (Fig. 1c). Overall, smell/taste changes are better correlated with the number of COVID-19 admissions to hospitals than the current governmental indicator i.e., the ratio of ER consultations for suspicion of COVID-19 to general ER consultations ($R_{smell} = 0.81$, $p = 6.71 \times 10^{-06}$ vs. $R_{gov} = 0.60$, $p = 3.35 \times 10^{-03}$); the same pattern was found for the number of COVID-19 related deaths ($R_{smell} = 0.75$, $p = 5.62 \times 10^{-05}$ vs. $R_{gov} = 0.58$, $p = 4.97 \times 10^{-03}$ see Supplementary Table 2). Further, when smaller geographical areas were considered (France is divided into 96 administrative units, called departments), these correlations remained highly significant (e.g., admissions to CCRUs: $R_{smell} = 0.76$, $p < 5 \times 10^{-19}$) (Fig. 1c). Moreover, the three relationships (change in smell/taste versus COVID-19-related hospitalization, resuscitations, and death) also remained highly significant when considering only individuals who were not clinically diagnosed by a medical professional but considering themself showing some symptoms of COVID-19 (e.g., admissions to CCRUs: $R_{smell} = 0.83$, $p = 1.65 \times 10^{-06}$). Potential sampling bias due to regional media coverage of our survey (Supplementary Table 3) and self-reported chemosensory changes by region was ruled out by confirming these variables were not correlated ($R < 0.01$, $p > 0.9$).

Notably, relationships between pandemic markers and online searches related to chemosensation were also significant in France. Google queries related to smell or taste loss ("perte odorat," "perte goût" in French) were correlated with the three measures of an overwhelmed healthcare system described above (e.g., CCRU admissions: $R_{smell} = 0.8$, $p < 4 \times 10^{-03}$, see Supplementary Table 2).

### Changes in smell and taste are early markers of the effectiveness of political decisions.
Next, we examined the temporal dynamics in France of self-reported changes in smell/taste, the current governmental indicator (ratio of ER consults), and the number of CCRU admissions due to COVID-19 before and after the lockdown period. As shown in Fig. 1d, the peak of the onset of changes in smell/taste appeared 4 days after the lockdown and for these individuals, the first reported COVID-19 symptoms occur even earlier. Conversely, the governmental indicator of ER consults only peaked 11 days after the lockdown, while the peak of CCRU admissions was shifted later by 14 days. This is consistent with emerging data showing that COVID-19-related changes in smell and taste occur in the first few days after infection[6,12–14]. The robustness of smell and taste changes over time was assessed in two ways. First, we showed the peak of smell/taste changes remained the same regardless of our survey's completion date (Supplementary Fig. 1a). Second, we observed the exact same peak when analyzing a separate French survey performed on 950 individuals and focusing on smell alterations in the French population independently of COVID-19 (see "Methods" section): the peak of olfactory changes again occurred 4 days after the lockdown decision, and this was independent of survey completion dates (Supplementary Fig. 1B). The robustness of smell and taste changes was also observed over age (Supplementary Fig. 2A) and gender (Supplementary Fig. 2B). Finally, we also show that the observed peak does not correspond to seasonal occurrence of allergies in France based on the ratio of consultations for Allergy to general consultations at the emergency room (Supplementary Fig. 3).

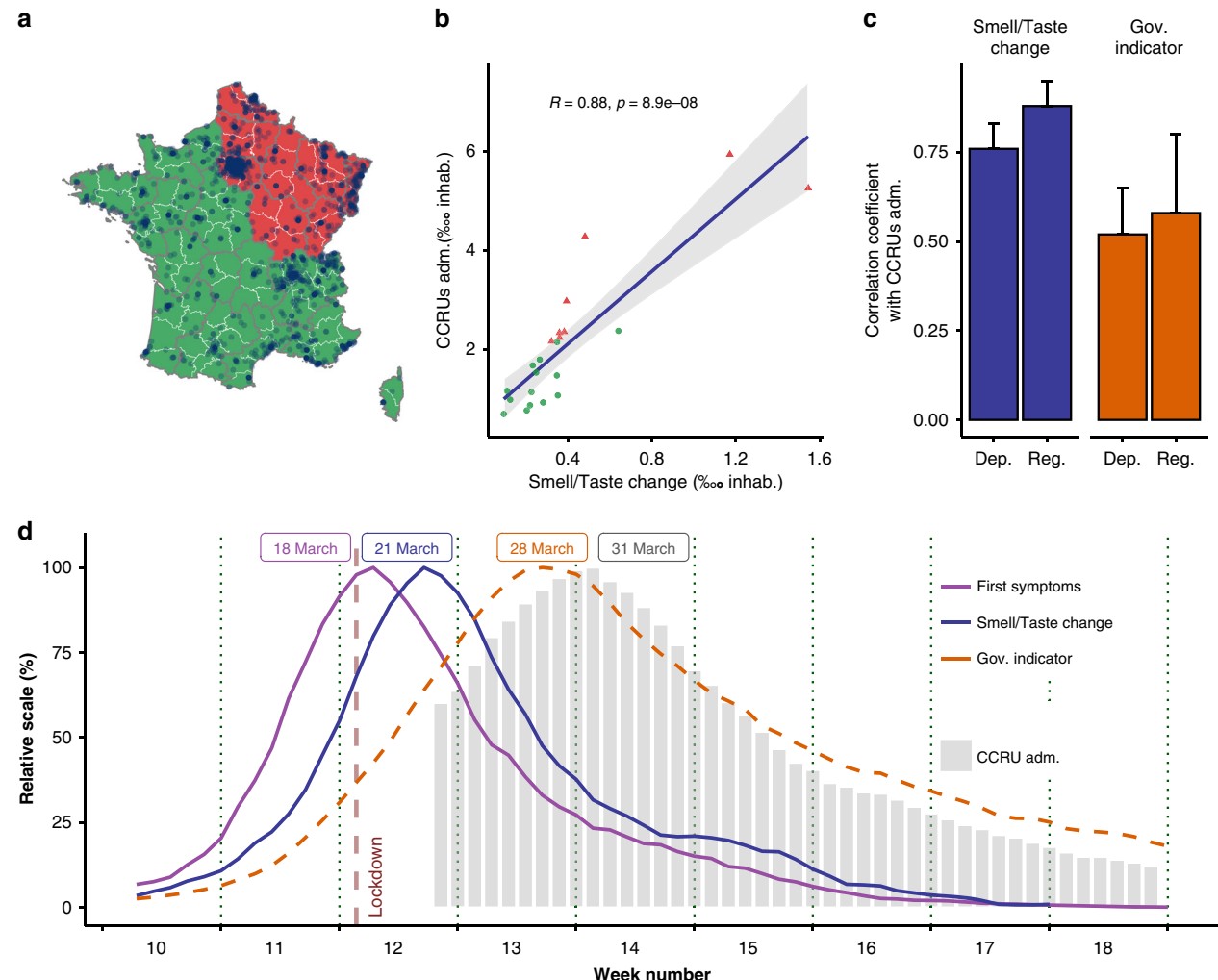

**Fig. 1 Changes in smell and taste as indicators of overwhelmed healthcare systems: geographic and time-related approaches. a** French regions were assigned a green or red status by the French government to guide local relaxation of lockdown protocols. Dots represent people self-reporting smell and taste changes in a web-based survey. Base map is from OpenStreetMap and OpenStreetMap Foundation. **b** The number of COVID-19-related CCRU admissions (as of May 11, 2020) correlated with the number of self-reported chemosensory changes (between March 1 and May 11, 2020, total $n = 3832$). Green dots correspond to regions with a post-lockdown level labeled green, and red triangles indicate regions considered red. Values are standardized based on the number of inhabitants (inhab.) for each regions. The two red triangles with CCRU admissions >5 are Alsace and Ile de France. The gray band represent the confidence interval of the linear smooth (formula 'y ~ x') $R$ and $p$ represent value of the test for association between paired samples, using one of Pearson's product moment correlation coefficient, without correction for multiple comparisons. **c** Colored bar represent the value of computed correlation coefficients (confidence intervals are depicted as thin black bars) between the number of CCRU admissions per area and i) the number of people reporting smell and taste changes ($n = 3832$, blue), and ii) the governmental indicator (Gov. indicator), ratio of ER consults for COVID-19 (orange). Analyses were done both at the level of metropolitan regions (Reg) and departments (Dep). **d** Temporal relationships in France between smell/taste change symptom onset (blue solid line, $n = 1476$), the governmental indicator (orange dashed line), and COVID-19 cases in CCRUs (gray bars) around the lockdown period (vertical dashed line). Data are 7-day running averages, normalized to the day with the highest value.

Further, analyses of Google searches confirm this temporal relationship: on the same days where survey participants report experiencing their first symptoms (around March 18, 2020), there was a peak of Google queries for terms associated with early COVID-19 symptoms (fever, cough, aches, Supplementary Fig. 4A). A few days later, the peak of online queries for "taste loss" and "smell loss" is synchronized with the report of smell and taste changes (Supplementary Fig. 4B). One week later, queries for shortness of breath preceded the peak of CCRU admissions (Supplementary Fig. 4C). Collectively, these results indicate a significant fraction of French COVID-19 patients followed the same symptom time course, experiencing initial symptoms at the very start of the lockdown, which might be representative of a peak of infection a few days before the lockdown. This is

consistent with the ultimate goal of the lockdown, which was to decrease the number of new infections following implementation. Thus, the period immediately prior to lockdown represents the expected peak of new infections. In France, a large population may have been infected two days before lockdown because that weekend was crowded and sunny and occurred over the course of election day. Further, there were busier train stations and supermarkets in anticipation of a shortage of supplies during lockdown[12].

These data suggest that the short-term efficacy of a lockdown could be monitored by tracking changes in smell and taste in the population. To assess whether such a prediction might generalize to other countries, we performed parallel analyses with data from Italy and the UK, where the lockdown measures were established

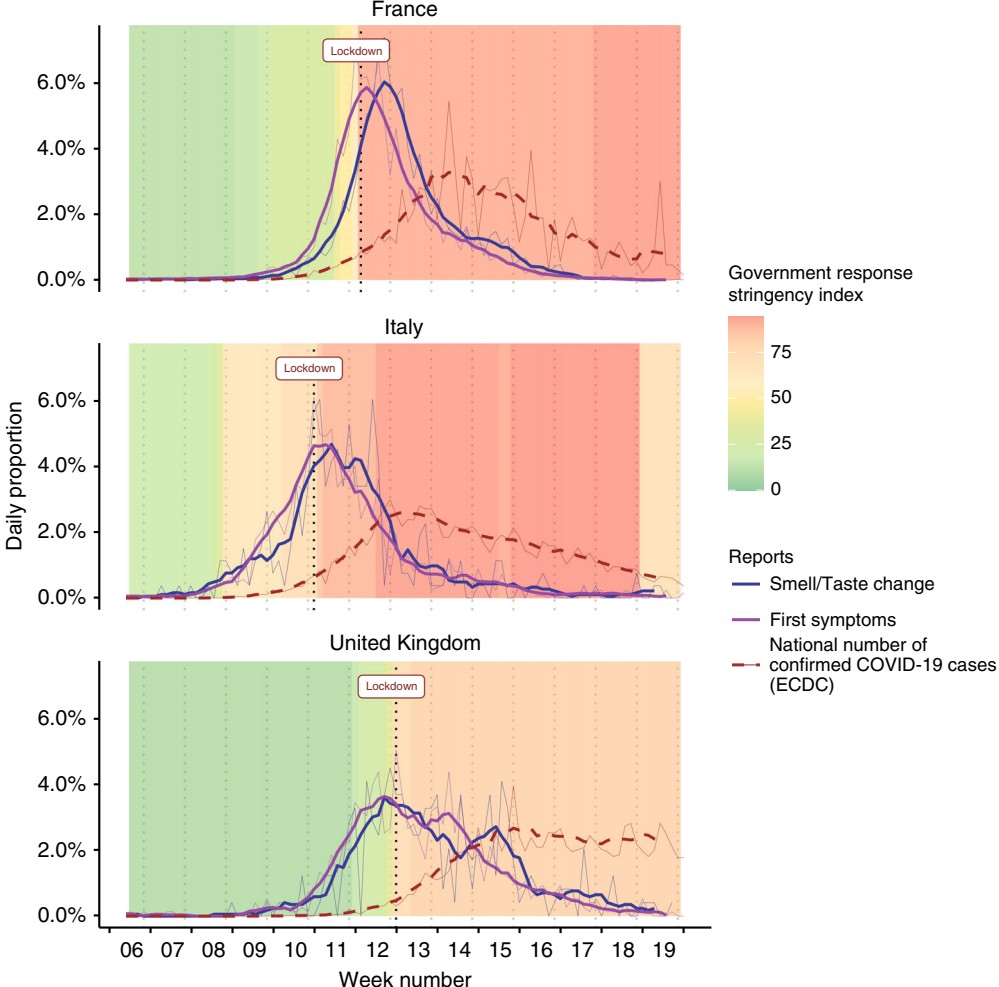

**Fig. 2 Evolution of COVID-19 indicators before and after the lockdown in France, Italy and the UK.** The daily proportion of first symptoms is shown as a violet line (France, n = 4720, Italy, n = 1241, UK, n = 750). The daily proportion of smell/taste changes is shown as a blue line (France, n = 1487, Italy, n = 264, UK, n = 263). The daily proportion of COVID-19 confirmed cases from the European Centre for Disease Prevention and Control (ECDC) is shown as a red dashed line. Each panel shows both raw data (thin line) and the corresponding 7-day running average (thick line). The government response stringency index is shown as the background color.

with different levels of severity (see Fig. 2). We monitored the dynamics of confirmed COVID-19 cases, self-reported first symptoms, and self-reported taste and smell changes, and compared them as a function of the governmental stringency index. Immediately after lockdown, we found that the two countries with the higher stringency index experienced a more rapid decrease in both self-reported smell and taste changes and COVID-19 symptoms. Further, as expected, the evolution of confirmed COVID-19 cases differs according to the stringency index. The governments of Italy and France rapidly increased their stringency index, which led to a sharp decrease in COVID-19 symptoms and cases. In contrast, in the UK, the number of people in the UK reporting symptoms showed a slower decrease, presumably due to a less severe lockdown policy, and the number of confirmed cases remained high during the observation window. In each country, self-reported smell and taste changes can be regarded as a useful metric to predict the dynamics of confirmed COVID-19 cases. That is, when the number of new onsets of chemosensory changes decreases sharply (France and Italy), the number of confirmed COVID-19 cases also decreases, albeit with a lag of two weeks. On the contrary, a slow decrease in the number of new onset chemosensory changes is associated with a plateau of confirmed cases (UK).

## Discussion

The present analyses reveal a strong spatial and temporal relationships between self-reported smell and taste changes and multiple indices of health care system stress, such as admissions to CCRUs. This is consistent with cumulative evidence showing a high prevalence of chemosensory alterations in patients affected by COVID-19 in Europe (France[8,14], Italy[9], UK[2,15,16]). Participants endorsed smell and taste changes only 3-4 days after their first symptoms. Such early chemosensory estimators may represent a cost-effective and easy way to implement alternative surveillance methods to large-scale virology tests, which are difficult to perform, costly, and time-consuming, especially during a pandemic.

A prominent question raised by these findings is whether the smell and taste changes observed in our study are solely related to COVID-19 or whether they can be explained by other temporal patterns, like seasonal illnesses or allergies. To the best of our knowledge, there are no existing studies that have explored the dynamics of sudden anosmia (as in COVID-19) throughout the year in France. Relationship between olfactory disturbances and seasons have been reported in Korea, Germany or US with a moderate increase of anosmia prevalence in spring[17–19]. Although the cyclical pattern of smell/taste changes might

overlap, the amplitude of reported changes (either due to allergy or viral affection) were very limited compared to the present report. To further rule out the possibility, we examined whether the annual peak of allergies in France could explain the peak of smell and taste changes observed here. In analyzing existing French governmental data, we found that the annual peak of allergies in France occurred around week 30 (beginning of summer), multiple weeks after the observation window of the present study (from week 5 to week 20, Supplementary Fig. 3). Further, the French national aerobiological surveillance network (RNSA, https://pollens.fr), which follows pollen concentration in the atmosphere, has also indicated the first week of lockdown was very low risk for seasonal allergies. In addition, when considering Google Trends data, we did not observe any similar peaks in queries for smell/taste loss in the corresponding time period in previous years. Finally, a comparative study in Israel[20] showed that in COVID-19 suspected patient the frequency of smell change is almost ten time higher in a COVID-19 positive patients (68%) than in COVID-19 negative (8%). Considering that most of the participants of the present study are diagnosed with COVID-19 and that their description of a sudden loss of smell/taste is consistent with the now typical presentation of COVID-19 symptoms, it is highly probable that COVID-19 infection is the main reason of their smell and taste change. Collectively, these data suggest the peak of smell and taste changes studied here are more consistent with sudden COVID-19 viral infections rather than an artifact due to seasonal illnesses.

The time lag between the onset of COVID-19-related symptoms and their declaration by the respondents of our study also deserves comment. Although immediate reporting of symptoms would have been ideal, such reporting is not possible within the context of the sudden first wave of a new viral pandemic. A similar time lag has been observed in other large-scale studies focusing on olfaction and COVID-19[21]. Indeed, this time lag is inevitable given the preparation time required for scientists and clinicians design and launch such a survey, with appropriate ethics approval, once anosmia and ageusia began to emerge as cardinal symptoms of COVID-19. The vast majority of participants completed the survey between April 10th and April 19th, 2020, and most of them declared a date of onset of their symptoms roughly a month earlier (although a small fraction of participants did indicate onset prior to 2020). A possible consequence of a time lag between survey completion and the effective date of symptom onset is that subjects' statements may have been influenced by major societal events such as the lockdown decision, potentially creating some recall bias. To examine whether the date of a major event like the lockdown might bias dates of reported smell and taste loss, we explored narrative descriptions provided by our participants. By analyzing responses to the optional open-ended question "Please describe the progression or order you noticed your symptoms", we observed that, for France, a mere 11 of 3705 people (who have filled the optional question) used the term "confinement" ("lockdown") in their description of the onset date. Separately, another factor that mitigates concerns about a potential recall bias is the stable nature of participant's statements, regardless of their date of completion. That is, logic suggests, the longer the time between the onset date of smell and taste loss and the reporting date, the greater the recall bias should be. However, our data clearly show that regardless of the date of completion, the onset date falls within the same period (Supplementary Fig. 1). Finally, other evidence against a potential recall bias comes from Google Trends data. Analyzing real-time Google queries in March, we observed a very particular trend in France (Supplementary Fig. 4). We first observed a peak of queries for terms associated with early COVID-19 symptoms (fever, cough, aches) synchronized with the declared onset of the first symptoms in the survey (around March 18th). A few days later, a peak of online queries for "taste loss" and "smell loss" was seen, and this was synchronized with the date reported of smell and taste changes in our survey. The striking concurrence between Google queries and reports in our survey argues against the idea that a recall bias could be driving the effects described here.

Another important factor to consider in our survey is the way the press and media might have influenced our findings. Indeed, when the survey was launched, smell and taste changes were reported as symptoms of COVID-19 in the national and local media, which might have influenced respondents to remind themselves of such symptoms and to then report these changes on the survey. Such an emphasis on smell and taste loss would have biased attempts to explore the prevalence of chemosensory deficits in COVID-19. However, the primary aim of the present investigation was not to focus on the prevalence of anosmia and ageusia with COVID-19, but rather to explore use of reported smell and taste loss as indicators of COVID-19 pandemic. Still, the media coverage of our survey could also have biased the selection of participants geographically, as some French regions received more media coverage than others. However, as reported above, there was no correlation between the number of participants in a given region and the intensity of media and press coverage for the survey in that same region. Finally, when participants were asked to describe the chronology of their symptoms, they did not refer to the media coverage as a prominent element influencing their awareness of their smell/taste changes. While this does not exclude an implicit and non-verbalized bias due to media coverage, this pattern suggests a genuine report of symptoms with a high occurrence of COVID symptoms just after the lockdown.

An interesting question raised by our findings is what impact they might have on government strategies in a pandemic. Following lockdown, the rapid decrease of self-reported changes in smell and taste in France may be representative of the effectiveness of this decision in reducing infection rates. Similarly, data from Italian participants show highly similar patterns, but with a one-week difference compared to the French data. This might reflect highly similar responses by the Italian and French governments. Conversely, the prevalence of chemosensory changes in the UK shows a more gradual decrease. The UK government began with advice to avoid pubs, clubs and theaters, and to work from home from March 16, with restrictions around March 18. However, a lockdown was not declared until March 23, and this was less stringent than those in France or Italy. Notably, new COVID-19 cases in the UK showed a plateau phase which is not observed in either France or Italy. Accordingly, we conclude that collecting online information about changes in smell and taste from residents (even retrospectively) may be a valuable metric of the effectiveness of reopening strategies related to the COVID-19 pandemic.

Practically, in areas where smell and taste changes are notable COVID-19 symptoms, the proportion of individuals who self-report changes in their ability to smell or taste might be an early indicator of subsequent demand for healthcare. If confirmed, continuous monitoring of changes in smell and taste perception would then be a highly cost-effective, minimally invasive, and reliable way to track future COVID-19 outbreaks. When used this way, we caution that particular attention must be paid to potential selection bias. That is, self-report studies online can be impacted by multiple selection biases, including socioeconomic status, fluency with technology and willingness and interest in participating in scientific research. When considering the present data, at least 3 parameters may contribute to a selection bias in our sample: (1) the age, (2) the gender of the participants, and (3)

the format and the advertising of the survey. Regarding participant' age, our study cohort (mean 40.7 years, sd = 12.4)) was quite similar to the French population mean (41.1 years, according to INSEE, https://www.insee.fr/fr/statistiques/1893198); however, we did only include individuals over 18 due to issues of consent, and administrative reasons, and seniors were also less represented. For gender, our sample contained a greater proportion of women (67%) compared to men, which might influence the results. However, additional analysis showed no differences in peaks of smell/taste changes across age or gender, minimizing concerns that such selection biases may have influenced present results (See Supplementary Fig. 2). We also tested the potential selection bias due to format and the advertising of the survey, by comparing the GCCR dataset with an independent second study performed on French residents (see "Methods" section). Remarkably we observed highly similar results across studies where advertising, inclusion criteria, and survey format were different.

Based on the present findings, we highlight the paramount importance and robustness of associations between smell/taste changes and COVID-19 and we strongly endorse the need for additional large-scale validation studies to assess the causality between the observed association between smell/taste changes and indicators of the COVID-19 pandemic. This could be achieved by setting up a simplified interface where selection biases are controlled for (age, gender, motivation, media coverage, socioeconomic level, etc.) through both traditional and online media—and whereby real time information about changes in smell and taste in the general population may be available to decision-makers. Subjects' participation in the questionnaire and the reliability of the answers should also be considered. In particular, if a participant knows how their answers may influence enforcement of lockdown, their answers might become less truthful. This motivation can be expressed through different forms of behavior. Whereas some individuals may tend to provide statements that minimize their symptoms in order to avoid strict containment measures, others will maximize their declaration to maintain the lockdown, or will provide honest answers in order to participate in the collective effort to better understand the COVID-19 pandemic. These motivational factors are a recurrent risk in online studies and different strategies should be held to control for them in future predictive studies. Based on the above, a large implementation of the study of smell and taste changes in institutional models should allow for monitoring of COVID-19 spread. This might be especially relevant in in areas in which testing proves difficult or delayed and for future outbreaks that may overlap with other seasonal viral diseases which share many of the symptoms (fever, cough etc.) but whose treatment or prevention (vaccination) are less demanding in terms of critical care than COVID-19. We advocate that self-report surveys should be used to enhance other strategies such as large-scale PCR tests and COVID-19 symptom assessments (including anosmia and ageusia) in primary/secondary care.

In summary, we propose that an increase in the incidence of sudden smell and taste change in the general population may be used as a valuable minimally invasive indicator of coronavirus spread in the population. To formally test the temporal relationship between chemosensory changes and spread of the disease, we recommend that a large-scale causal study in different countries be conducted on real-time monitoring of self-reported changes in the ability to smell or taste. Such a prospective study will allow for the creation of statistical models that can assist in prediction of future hospital admissions for COVID-19. Further, it could also help decision-makers take important measures at the local level, either in catching new outbreaks sooner, or in guiding the relaxation of local lockdowns, given the strong impact of lockdown on economic and social activities.

## Methods

**Online survey**. This study is mainly based on data from the Global Consortium for Chemosensory Research survey (GCCR, https://gcchemosensr.org/) – a global, crowd-sourced online study deployed in 30+ languages[22]. The data analyzed here were collected from April 7 to May 14, 2020. The protocol complies with the revised Declaration of Helsinki and was approved as an exempt study by the Office for Research Protections at The Pennsylvania Study University (Penn State) in the U.S.A. (STUDY00014904; PI Hayes).

Participants in the GCCR questionnaire were recruited by word of mouth, as well as through social and traditional media (flyers, social media, television, radio) during the COVID-19 pandemic. It was well covered by the French press, as over 70 articles mentioned the project, at both the regional and national level (see Supplementary Table 3). Respondents received no monetary incentive for their participation. Inclusion criteria were as follows. (i) Questionnaire completion was allowed only to participants who indicated they had suffered from a respiratory disease in the past two weeks, whether they noticed a change in their taste/smell or not. (ii) Participants aged 18 years old or younger were excluded.

For the analyses conducted in this article, only individuals reporting a change in smell and/or taste perception were included, based on the question "Have you had any of the following symptoms with your recent respiratory illness or diagnosis?". Moreover, to exclude unreliable entries, participants must have reported a quantitative decrease of at least 5 on a 0-to-100 rating scale between their ability to smell and/or taste before and during their recent respiratory illness or diagnosis. Therefore, Due to this inclusion criteria, "smell/taste change" is equivalent to a quantitative decrease of participant ability to smell and/or taste. We then extracted individuals from the full dataset who reported living in France, Italy or the UK. As the country of residence was completed as a text entry, we allowed for typical variations (e.g., "United Kingdom" or "UK"), spelling mistakes, use of different languages (e.g., "Italie" or "Italia"), as well as subdivisions (e.g., "Scotland") and major cities ("Paris"). Metropolitan France was split into 13 so-called "regions" in 2016. However, we considered the former system where France was split into 22 regions here, since the organization of the health system mostly remains based on the structure built before 2016. An alternative, finer granularity, splits metropolitan France into 96 so-called "departments." To retrieve the French department and region of the participants, we used the city of residence they reported in the questionnaire and combined them with the French public website (data.gouv.fr, after a semi-manual correction of spelling). Participants came from all metropolitan departments but three (Mayenne, Creuse, Cantal). Consequently, the number of responses analyzed in France was between $n = 1476$ and 4720 depending on the analysis conducted (i.e., on whether the information of interest was present or missing and the date range of analysis, see Supplementary Table 1 for details). For comparison, between 264 to 1241 participants from Italy and between 243 to 750 participants from the UK were included. Most participants were women (FR:66.38%, IT:69.3%, UK:76.0%), and the mean age was around 40 [FR = 40.7 (sd = 12.4), IT = 41.1 (sd = 11.4), UK = 41.09 (sd = 12.1)]. In the French data, a total of 15% of individuals tested positive for COVID-19 (lab result) and 44% were diagnosed clinically by a medical professional from their symptoms. The remaining 41% were not diagnosed for COVID-19 but declared a change in perception of either smell or taste. The number of participants was normalized by region, by using the number of inhabitants in each region as estimated by the French public statistics office, INSEE. Finally, the time of onset of smell and taste change was assessed via responses to several optional open-ended questions. These included: "Please describe the progression or order you noticed your symptoms" and the time of onset of recent disease by the question: "At what date did you first notice symptoms of your recent respiratory illness? Provide your best guess or leave blank if you do not remember."

**Complementary and independent French Survey**. The data of another online survey were used to evaluate the robustness of the temporal evolution of smell and taste changes. This survey was conducted in the French population between April 8 and May 8, 2020 and aimed at characterizing chemosensory disorders in people with and without COVID-19, as well as their consequences on quality of life. The data of 950 respondents were eligible for comparison with data from the GCCR survey, i.e., responses where both the date of completion and the date of smell loss onset were provided. Only responses that were complete and from people who were responding to the questionnaire for the first time and were over age 18 were included. This survey was approved by the CNRS ethics committee. Data collection was strictly anonymous. The protocol complies with the revised Declaration of Helsinki and the study was approved by the ethics committee of the Institute of Biological Sciences of the CNRS on the 3rd of April 2020 (DPO #TRRECH-467). All individuals provided informed consent when participating in the survey.

**Online trends**. Trends of online queries by French region were performed using Google Trends, a tool returning the popularity of a search term in a specific state or region. Google is by far the most used search engine in France (>90% of internet searches, according to StatCounter Global Stats). We looked for the popularity of terms (listed in Supplementary Fig. 3, using default selection of "All categories" and "Web search"), within the timeframe of February 1, 2020 to May 10, 2020 (from the month of the first official COVID-related death in Europe to the end of lockdown in France). It should be noted that Google Trends does not provide the

actual numbers of searches but rather a relative score from 0 to 100 (100 corresponding to the day with the greatest number of searches during the specified time period). To compare Google Trends scores between French regions, we transformed them by computing the relative number of queries per day in the region of interest. For example, despite a value of 100, the peak day might represent only 5% of the total number of queries related to the topic across the timeframe of interest (see above).

**Healthcare system data**. The French governmental indicator to estimate the circulation of the virus was calculated from the ratio of consultations for suspected COVID-19 to general consultations at the emergency room (ER) in hospitals. This ratio corresponds to the medical diagnostic for COVID-19 suspicion (codes CIM10: U07.1, U07.10, U07.11, U07.12, U07.14, U07.15, U04.9, B34.2, B97.2). The definition of COVID-19 has evolved rapidly during the lockdown period but the diagnosis is principally based on symptoms of COVID-19 considered as common such as fever, cough, and dyspnea (difficulty breathing). To the best of our knowledge, anosmia and ageusia were officially considered in France as putative symptoms of COVID-19 from a letter of the Direction Générale de la Santé (April 1st) and communication of the Haut Conseil de la Santé Publique (a letter dated April 8, published online April 15, following a letter from the CNP-ORL dated March 20). Areas with values of the French governmental indicator higher than 10% are considered having a high virus circulation. This indicator contributes to the assignment of a red/green label. Allergies incidence in previous years were calculated from the ratio of consultations for Allergy to general consultations at the emergency room (ER) in hospitals.

Data dealing with the health status across countries (number of COVID-19 cases and deaths for each day) were downloaded on May 22, 2020 from the European Centre for Disease Prevention and Control databank (ECDC, https://www.ecdc.europa.eu/en). Data regarding healthcare system stress in France (hospitalizations, CCRU entries and deaths) were also downloaded on May 22 from the French Public Health website (Géodes, Santé Publique France, https://geodes.santepubliquefrance.fr/#c=home). Here, we use the term CCRU (Critical Care Resuscitation Unit) to translate the French hospital service of "Réanimation." Raw data were normalized across regions with regard to their number of inhabitants as estimated by INSEE. The temporal evolution of the stringency of government response was retrieved from the Oxford COVID-19 (https://www.bsg.ox.ac.uk/research/research-projects/coronavirus-government-response-tracker). Here, the stringency level of a country is computed according to which measures of a list of items (e.g., school closures, cancellation of public events, international travel controls, etc.) are undertaken. For the post-lockdown situation, the color assigned by the French government to each department was downloaded on May 12 from the government website. Only data before May 11 (the initial lift of the lockdown) were included in the analyses.

**Statistical analyses**. Statistical analyses were pre-registered at the Open Science Framework (OSF). Data were analyzed using R software (4.0) and its standard packages (maps, ggplot, etc.). Data were grouped at the national level (France, Italy, UK). In France they were also grouped at the regional level (according to the division into 22 regions in place prior to the 2016 reform). The rationale behind this is that the healthcare system is still structured following this organization, with University Hospitals in regional main cities serving patients of the surrounding departments. Participants from overseas French territories were not included in the geographical analysis because of too few data ($n < 10$). The relationship between (1) GCCR responses (or online queries), and (2) public health data was determined using parametric (e.g., Pearson correlations) statistics as allowed by the normal distribution of the variable of interest. The association between GCCR participant and red/green post-lockdown status was tested using Chi-square tests and Biserial correlations. Complementary analyses not planned in the pre-registration included: (i) the analysis using the independent French online survey (see section "Complementary and independent French Survey" of the methods), (ii) the correlation between regional media coverage and the number of responses to the online survey per region, (iii) the correlation at the level of department, (iv) the correlation excluding extreme points, and (v) the correlation with the government indicator. Pre-registered statistical analyses not presented here include: (i) Mann-Kendall trend test and Change-point Detection test to detect time series changes, and (ii) part of the Google Trends analysis.

**Reporting summary**. Further information on research design is available in the Nature Research Reporting Summary linked to this article.

## Data availability
The authors declare that the data supporting the findings of this study are available within the paper and its supplementary information files. (Source Data file). Source data are provided with this paper.

## Code availability
R scripts are available on the osf server (https://osf.io/gew7p/).

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

## Acknowledgements
This work was supported financially by EXTREM-O (CNRS MITI), CONFINEZ2 (CNES), the CORODORAT grant (IDEX-Lyon-Université de Lyon), by the French government, through the UCAJEDI "Investments in the Future" project managed by the ANR grant No. ANR-15-IDEX-01. The authors thank Sébastien Buthion, Conceição Silva, Brigitte Perucca, Clément Blondel, Stéphanie Younès, François Maginot, and their teams from the CNRS communication units, as well as the Global Consortium for Chemosensory Research (GCCR). Deployment of the GCCR survey in multiple languages by the Pennsylvania State University (Penn State) was partially supported by funds from James and Helen Zallie and the Penn State Sensory Evaluation Center.

## Author contributions

Conceived, designed the study, and wrote the paper: D.P., V.P.-L., C.F., J.G., M.B. Data acquisition and curation: D.P., V.P., M.M., M.M., E.B., O.A., J.K., M.H., J.P., R.D., C.D., S.S., E.M., M.C.F., K.W.C., E.A.S., T.T.D., A.J.B., V.P., J.E.H., T.L., C.F., J.G., M.B. Performed analysis: D.P., V.P.-L., C.F., J.G., M.B., M.M.a., M.M.o., E.B. Edited and approved the final manuscript: D.P., V.P., M.M., M.M., E.B., O.A., J.K., M.H., J.P., R.D., C.D., S.S., E.M., M.C.F., K.W.C., E.A.S., T.T.D., A.J.B., V.P., J.E.H., T.L., C.F., J.G., M.B.

## Competing interests

The authors declare no competing interests.
