## [Peer Review File · Nature Communications]

REVIEWER COMMENTS

Reviewer #1 (Remarks to the Author):

The authors propose a novel method for anticipating excess workload in critical care during the (or potentially further waves of the) COVID pandemic. A self-reported study is provided with around 5,000 respondents.

The statistical analyses presented consist mainly in chi-squared tests and tests of correlational association. However, proposed changes in detection/anticipation at a healthcare-system level would need causal evidence. This paper does not provide evidence at that level - it is observational. Moreover selection and recall bias may be present in the study (the authors provide one test of selection bias due to media coverage) and the authors do not explore this issue in detail.

The authors provide a well-written manuscript and present, it seems to this reviewer, a case for a linked primary-secondary care study of hospital presentations for respiratory complaints with clear temporal evidence of events. Perhaps even an in-hospital study.

Reviewer #2 (Remarks to the Author):

"Self-reported smell and taste changes are early indicators of the COVID-19 pandemic and of the effectiveness of political decisions" by Dr Pierron and colleagues.

The data presented is interesting as it suggests that the local epidemiology of COVID-19 can be monitored by tracking changes in smell and taste in the population. Such an early warning system may be useful to evaluate the effects on different lockdown and reopening strategies in the current COVID-19 pandemic.

The manuscript is well written the data is clearly illustrated in figures. Survey data on smell and/or taste changes from France and, to a lesser extent, Italy and the UK, too, appear to closely follow and anticipate the number of verified cases of COVID-19. In France, the number of cases that indicate smell and/or taste changes is preceded by the number of admissions to intensive care by 1-2 weeks. Such warning 1-2 week in advance of an increase in the burden on the healthcare system in one area due to COVID-19 is of potentially exceptional value to health planners and stakeholders.

However, I do have some considerations and concerns regarding the generalizability of the model and the results when it comes to its ability to in real-time monitor the epidemiology of COVID-19 in different countries. First, the data presented seems to represent self-reported chemosensory changes that occurred between March 1 and May 11 but were reported in retrospect during April 7 to May 14, i.e at least 3 weeks after the lockdown in France 17 March. The authors need a clarify the time lag between when the COVID-19 related symptoms occurred in the respondents and when the symptoms were reported.

Second, when the study was conducted, the remarkable COVID-19 symptom profile with taste and smell changes was frequently reported and noted in news media. This may have influenced individuals' tendency to remind themselves and also to report these observations via the survey.

Third, the lock down had already been implemented when the data was collected. Thus, the potential disadvantage of reporting previous occurrence of COVID-19 related symptoms probably appeared insignificant for the participants on the grounds that this, in turn, would increase the risk

of further restrictive measures by the authorities.

As the ramifications of different lock-down strategies will have obvious and far-reaching negative effects on society at large but also on individuals' opportunities to socialize with each other, support themselves and their family, and educational opportunities, this may adversely affect individuals' inclination to report COVID-19 related symptoms based on a concern that the authorities thus may reintroduce different lock-down measures.

The authors need to discuss these inherent limitations and reflect on what the impact and significance they will have the conclusion of the study. To show that the method is excellent to track and monitor the occurrence of COVID-19 related symptoms in retrospect is one thing. To imply that it may be a valuable indicator of the effectiveness of reopening strategies related to the COVID-19 is something completely different.

Micael Widerström

Reviewer #3 (Remarks to the Author):

The study by Pierron et al. utilized multiple data sources, from a chemosensory survey, online search, government indicators and COVID admissions counts to demonstrate the use of smell/taste loss as an early indicator of COVID-19, well before hospital admissions and suspected COVID-19 cases. The analyses were carefully performed and seems robust. There is a potential for a low-cost syndromic surveillance system.

Major comments

1. The authors have carefully analyzed the data and results seem to be robust.
2. Methods, Online survey. The GCCR data were analyzed from April 7 to May 14. Figure S1 shows GCCR data since mid-February. Does it mean that it involved retrospective reporting up to 2 months or more?
3. Figure S1 legend. Following the above questions, am I correct that 'Before Apr 20' was actually 'Apr 7-19' and 'Before Apr 13' was actually 'Apr 7-12'?
4. Methods, Online survey. Have the authors considered the direction of change in the rating of smell and/or taste, and considered loss of smell/taste (e.g. decrease in rating by 5 marks or more)?
5. Methods, healthcare system data. The government indicator used suspected COVID-19 to general consultations at the ER. Please state the definition of the suspected COVID-19 in France. Did the definition related to taste or smell?
6. Would the authors recommend change in the case definition for suspected or probable COVID-19 cases with regard to taste or smell loss?
7. This is the main criticism of the study. Was there any seasonal pattern in the taste/smell loss in the previous years? For example, a seasonal pattern every year around week 10-12 in France would provide an alternative explanation of observations.
8. This is not essential but would strengthen the study – If it can be demonstrated that smell/taste loss indicator can show early signal of resurgence (e.g. in Sweden).

Reviewer #1 Remarks :

1 >> *"The statistical analyses presented consist mainly in chi-squared tests and tests of correlational association. However, proposed changes in detection/anticipation at a healthcare-system level would need causal evidence. This paper does not provide evidence at that level - it is observational. "*

We agree with the reviewer that public health decisions should be based on a strong predictive model - with causality - with also a broader data collection system (that only health institutions can develop). Accordingly, in the revised version, we emphasize multiple times that this work is correlational in nature, and needs to be validated in follow-up studies to determine causality before policy changes are enacted. Likewise, we have also revised the conclusion.

- **Change in DISCUSSION (p.13, l.0):** *"Based on the present findings, we highlight the paramount importance and robustness of associations between smell/taste changes and COVID-19 and we strongly endorse the need for additional large-scale validation studies to assess the causality between the observed association between smell/taste changes and indicators of the COVID-19 pandemic. This could be achieved by setting up a simplified interface where selection biases are controlled for (age, gender, motivation, media coverage, socioeconomic level etc.) through both traditional and online media – and whereby real time information about changes in smell and taste in the general population may be available to decision-makers. "*
- **Change in DISCUSSION - Last paragraph (p.14, l.7):** *"In summary, we propose that an increase in the incidence of sudden smell and taste change in the general population may be used as a valuable minimally invasive indicator of coronavirus spread in the population. To formally test the temporal relationship between chemosensory changes and spread of the disease, we recommend that a large-scale causal study in different countries be conducted on real-time monitoring of self-reported changes in the ability to smell or taste. Such a prospective study will allow for the creation of statistical models that can assist in prediction of future hospital admissions for COVID-19. Further, it could also help decision-makers take important measures at the local level, either in catching new outbreaks sooner, or in guiding the relaxation of local lockdowns, given the strong impact of lockdown on economic and social activities."*

2 >> *"Moreover selection and recall bias may be present in the study (the authors provide one test of selection bias due to media coverage) and the authors do not explore this issue in detail."*

We thank the reviewer for raising this excellent point -- in the revised version, we address this concern in detail. First, regarding selection biases, a series of three parameters were considered: the survey format and age & gender of the participants. Second, with regard to potential recall biases, we agree with the fact that the participants' statements may have been biased by specific past events such as governmental lockdown decisions. To control for this,

we compared the two independent studies and showed that the peak is independent of the date of completion. We also performed an additional analysis on open ended verbatim comments from the participants, which showed that very few mentioned events that may have influenced their declaration (i.e. lockdown...). We included all these new analyses and these limitations in the revised version.

- **recall bias- Change in DISCUSSION (p.10, l.18) :** *”A possible consequence of a time lag between survey completion and the effective date of symptom onset is that subjects’ statements may have been influenced by major societal events such as the lockdown decision, potentially creating some recall bias. To examine whether the date of a major event like the lockdown might bias dates of reported smell and taste loss, we explored narrative descriptions provided by our participants. By analyzing responses to the optional open-ended question “Please describe the progression or order you noticed your symptoms”, we observed that, for France, a mere 11 of 3705 people (who have filled the optional question) used the term “confinement” (“lockdown”) in their description of the onset date. Separately, another factor that mitigates concerns about a potential recall bias is the stable nature of participant’s statements, regardless of their date of completion. That is, logic suggests, the longer the time between the onset date of smell and taste loss and the reporting date, the greater the recall bias should be. However, our data clearly show that regardless of the date of completion, the onset date falls within the same period (**Supplementary figure S1**). Finally, other evidence against a potential recall bias comes from Google trend data. Analyzing real-time Google queries in March, we observed a very particular trend in France (**Supplementary figure S4**). We first observed a peak of queries for terms associated with early COVID-19 symptoms (fever, cough, aches) synchronized with the declared onset of the first symptoms in the survey (around March 18). A few days later, a peak of online queries for “taste loss” and “smell loss” was seen, and this was synchronized with the date reported of smell and taste changes in our survey. The striking concurrence between Google queries and reports in our survey argue against the idea that a recall bias could be driving the effects described here.”*
- **selection bias - Change in RESULT (p.6, l.15):** *“The robustness of smell and taste changes was also observed over age (Supplementary Figure 2A) and gender over gender (Supplementary Figure 2B). “*
- **selection bias - Change in DISCUSSION (p.12, l.18):** *“If confirmed, continuous monitoring of changes in smell and taste perception would then be a highly cost-effective, minimally invasive, and reliable way to track future COVID-19 outbreaks. When used this way, we caution that particular attention must be paid to potential selection bias. That is, self-report studies online can be impacted by multiple selection biases, including socioeconomic status, fluency with technology and willingness and interest in participating in scientific research. When considering the present data, at least 3 parameters may contribute to a selection bias in our sample: (1) the age, (2) the gender of the participants, and (3) the format and the advertising of the survey.*

Regarding participant' age, our study cohort (mean 40.7 yr, $sd=12.4$) was quite similar to the French population mean (41.1 yr, <https://www.insee.fr/fr/statistiques/2381476>); however, we did only include individuals over 18 due to issues of consent, and administrative reasons, and seniors were also less represented. For gender, our sample contained a greater proportion of women (67%) compared to men, which might influence the results. However, additional analysis showed no differences in peaks of smell/taste changes across ages or gender, minimizing concerns that such selection biases may have influenced present results (See **Supplementary Figure 2**). We also tested the potential selection bias due to format and the advertising of the survey, by comparing the GCCR dataset with an independent second study performed on French residents (see Methods). Remarkably we observed highly similar results across studies where advertising, inclusion criteria, and survey format were different”

- **selection bias - Supplementary Figure 2 (p25, l.3):** “Time of peaks of smell/taste changes in French participants who answered the GCCR questionnaire according to their age or gender. Similarly, to figure S1, colored lines represent the proportion of reported onset of smell and taste normalized to the highest value of each series. Colors represent groups of participants according to age (S2.A) or gender (S2.B). “

Reviewer #2 Remarks :

However, I do have some considerations and concerns regarding the generalizability of the model and the results when it comes to its ability to in real-time monitor the epidemiology of COVID-19 in different countries.

>> First, the data presented seems to represent self-reported chemosensory changes that occurred between March 1 and May 11 but were reported in retrospect during April 7 to May 14, i.e at least 3 weeks after the lockdown in France 17 March. The authors need to clarify the time lag between when the COVID-19 related symptoms occurred in the respondents and when the symptoms were reported.

We agree with the reviewer; this point has been considered and clarified in the new version of the discussion. In addition, we also discuss the recall bias (see answer to comment 2 of reviewer 1).

- **Change in DISCUSSION (p.10, l.10) :** "The time lag between the onset of COVID-19-related symptoms and their declaration by the respondents of our study also deserves comment. Although immediate reporting of symptoms would have been ideal, such reporting is not possible within the context of the sudden first wave of a new viral pandemic. A similar time lag has been observed in other large-scale studies focusing on olfaction and COVID-19. Indeed, this time lag is inevitable given the preparation time required for scientists and clinicians design and launch such a survey, with appropriate ethics approval, once anosmia and ageusia began to emerge as cardinal symptoms of COVID-19. The vast majority of participants completed the survey between April 10th and April 19th 2020, and most of them declared a date of onset of their symptoms roughly a month earlier (although a small fraction of participants did indicate onset prior to 2020)."

>> Second, when the study was conducted, the remarkable COVID-19 symptom profile with taste and smell changes was frequently reported and noted in news media. This may have influenced individuals' tendency to remind themselves and also to report these observations via the survey.

We thank the reviewer for this comment, we added a discussion regarding the impact of the media.

- **Change in DISCUSSION (p.11, l.13) :** "Another important factor to consider in our survey is the way the press and media might have influenced our findings. Indeed, when the survey was launched, smell and taste changes were reported as symptoms of COVID-19 in the national and local media, which might have influenced respondents to remind themselves of such symptoms and to then report these changes on the survey. Such an emphasis on smell and taste loss would have biased attempts to explore the prevalence of chemosensory deficits in COVID-19. However, the primary aim of the present investigation was not to focus on the prevalence of anosmia and ageusia with COVID-19, but rather to explore use of reported smell and taste loss as indicators of

COVID-19 pandemic. Still, the media coverage of our survey could also have biased the selection of participants geographically, as some French regions received more media coverage than others. However, as reported above, there was no correlation between the number of participants in a given region and the intensity of media and press coverage for the survey in that same region. Finally, when participants were asked to describe the chronology of their symptoms, they did not refer to the media coverage as a prominent element influencing their awareness of their smell/taste changes. While this does not exclude an implicit and non-verbalized bias due to media coverage, this pattern suggests a genuine report of symptoms with a high occurrence of COVID symptoms just after the lockdown “

>> Third, the lock down had already been implemented when the data was collected. Thus, the potential disadvantage of reporting previous occurrence of COVID-19 related symptoms probably appeared insignificant for the participants on the grounds that this, in turn, would increase the risk of further restrictive measures by the authorities. As the ramifications of different lock-down strategies will have obvious and far-reaching negative effects on society at large but also on individuals' opportunities to socialize with each other, support themselves and their family, and educational opportunities, this may adversely affect individuals' inclination to report COVID-19 related symptoms based on a concern that the authorities thus may reintroduce different lock-down measures.

*In all large-scale studies in which participants report symptoms, motivational biases are unfortunately inevitable. And we agree with the reviewer that if the potential disadvantage of reporting previous occurrence of COVID-19 related symptoms probably appeared insignificant for the participants it should not be ruled out for further study. Also, like many affected countries, the COVID-19 pandemic in France resulted in a great surge of solidarity in all strata of society, ranging from the preparation of meals for caregivers by ordinary citizens to the manufacture of masks for all trades mobilized during containment. This said, even if the hypothesis of "frank" participation in an international survey to better understand COVID-19 is not to be ruled out, this point has to be studied in order to assess the *the generalizability of the model and the results when it comes to its ability to monitor in real-time the epidemiology of COVID-19 in different countries.**

- **Change in DISCUSSION (p.13, l.16) :**” *Subjects’ participation in the questionnaire and the reliability of the answers should also be considered. In particular, if a participant knows how their answers may influence enforcement of lockdown, their answers might become less truthful. This motivation can be expressed through different forms of behavior. Whereas some individuals may tend to provide statements that minimize their symptoms in order to avoid strict containment measures, others will maximize their declaration to maintain the lockdown, or will provide honest answers in order to participate in the collective effort to better understand the COVID-19 pandemic. These motivational factors are a recurrent risk in online studies and different strategies should be held to control for them in future predictive studies.”*

The authors need to discuss these inherent limitations and reflect on what the impact and significance they will have the conclusion of the study. To show that the method is excellent to track and monitor the occurrence of COVID-19 related symptoms in retrospect is one thing. To imply that it may be a valuable indicator of the effectiveness of reopening strategies related to the COVID-19 is something completely different.

We agree with the reviewer, we added a significant part of the discussion over existing challenges before proposing a real-time *study* and changed the conclusions.

- **Change in DISCUSSION (p.13, l.10)** “Based on the present findings, we highlight the paramount importance and robustness of associations between smell/taste changes and COVID-19 and we strongly endorse the need for additional large-scale validation studies to assess the causality between the observed association between smell/taste changes and indicators of the COVID-19 pandemic. “
- **Change in DISCUSSION last paragraph (p.14, l.7):** In summary, we propose that an increase in the incidence of sudden smell and taste change in the general population may be used as a valuable minimally invasive indicator of coronavirus spread in the population. To formally test the temporal relationship between chemosensory changes and spread of the disease, we recommend that a large-scale causal study in different countries be conducted on real-time monitoring of self-reported changes in the ability to smell or taste. Such a prospective study will allow for the creation of statistical models that can assist in prediction of future hospital admissions for COVID-19. Further, it could also help decision-makers take important measures at the local level, either in catching new outbreaks sooner, or in guiding the relaxation of local lockdowns, given the strong impact of lockdown on economic and social activities.
-

Reviewer #3 Remarks :

>> *Methods, Online survey. The GCCR data were analyzed from April 7 to May 14. Figure S1 shows GCCR data since mid-February. Does it mean that it involved retrospective reporting up to 2 months or more?*

We clarified the time lag in the discussion. (cf and comment 1 of reviewer 2).

- **Change in DISCUSSION (p.10, l.10):** “The time lag between the onset of COVID-19-related symptoms and their declaration by the respondents of our study also deserves comment. Although immediate reporting of symptoms would have been ideal, such reporting is not possible within the context of the sudden first wave of a new viral pandemic. A similar time lag has been observed in other large-scale studies focusing on olfaction and COVID-19. Indeed, this time lag is inevitable given the preparation time required for scientists and clinicians design and launch such a survey, with appropriate ethics approval, once anosmia and ageusia began to emerge as cardinal symptoms of COVID-19. The vast majority of participants completed the survey between April 10th and April 19th 2020, and most of them declared a date of onset of

their symptoms roughly a month earlier (although a small fraction of participants did indicate onset prior to 2020).”

>> *Figure S1 legend. Following the above questions, am I correct that ‘Before Apr 20’ was actually ‘Apr 7-19’ and ‘Before Apr 13’ was actually ‘Apr 7-12’?*

We clarified the *Figure S1 legend*. (‘Before Apr 20’ = ‘Apr 7-19’ ; ‘Before Apr 13’ = ‘Apr 8-12’)

>> *Methods, Online survey. Have the authors considered the direction of change in the rating of smell and/or taste, and considered loss of smell/taste (e.g. decrease in rating by 5 marks or more)?*

The reviewer is correct, we clarified it in the method section.

- **METHODS (p.17, l.16):** “For the analyses conducted in this article, only individuals reporting a change in smell and/or taste perception were included, based on the question “Have you had any of the following symptoms with your recent respiratory illness or diagnosis?”. Moreover, to exclude unreliable entries, participants must have reported a quantitative decrease of at least 5 on a 0-to-100 rating scale between their ability to smell and/or taste before and during their recent respiratory illness or diagnosis. Therefore, due to this inclusion criteria, “smell/taste change” is equivalent to a quantitative decrease of participant ability to smell and/or taste. “

>> *Methods, healthcare system data. The government indicator used suspected COVID-19 to general consultations at the ER. Please state the definition of the suspected COVID-19 in France. Did the definition related to taste or smell?*

We clarified this point in the method section.

- **METHODS (p.19, l.22):** “The French governmental indicator to estimate the circulation of the virus was calculated from the ratio of consultations for suspected COVID-19 to general consultations at the emergency room (ER) in hospitals. This ratio corresponds to the medical diagnostic for COVID-19 suspicion (codes CIM10 : U07.1, U07.10, U07.11, U07.12, U07.14, U07.15, U04.9, B34.2, B97.2). The definition of COVID-19 has evolved rapidly during the lockdown period but the diagnosis is principally based on symptoms of COVID-19 considered as common such as fever, cough, and dyspnea (difficulty breathing). To the best of our knowledge, anosmia and ageusia were officially considered in France as putative symptoms of COVID-19 from a letter of the Direction Générale de la Santé (1st April) and communication of the Haut Conseil de la Santé Publique a (letter dated 8 April, published online 15 April, following a letter from the CNP-ORL dated 20 March).”

>> *Would the authors recommend change in the case definition for suspected or probable COVID-19 cases with regard to taste or smell loss?*

The definition of COVID-19 has evolved rapidly during the last months and taste or smell loss are presently considered as a symptom for suspected or probable COVID-19 cases in France.

In addition, we recommend that smell/taste loss incidence at the population level should be considered as an indicator of COVID-19 spread.

7. This is the main criticism of the study. Was there any seasonal pattern in the taste/smell loss in the previous years? For example, a seasonal pattern every year around week 10-12 in France would provide an alternative explanation of observations.

To the best of our knowledge there is no epidemiological dataset allowing to directly assess seasonal patterns of anosmia and ageusia in France. Alternatively, we explored seasonal patterns that - beside viral infection - can provoke anosmia, namely respiratory allergies. In addition to the reviewer's comment, we also reported the time and geography of allergic risk to pollen in France in 2020 during the lockdown. This shows that the peak of reported changes in smell and taste does not correspond to the peak of allergic risk in the concerned regions.

- **METHODS (p.20, l.7):** “Allergies incidence in previous years were calculated from the ratio of consultations for Allergy to general consultations at the emergency room (ER) in hospitals.”
- **RESULTS (p6, l.15):** “Finally, we also show that the observed peak does not correspond to seasonal occurrence of allergies in France based on the ratio of consultations for Allergy to general consultations at the emergency room (Supplementary Figure 3).“
- **Figure S3 (p26, l.1):** “Evolution of the ratio of consultation for allergies in the emergency room (for 10.000 consultation) over time during the last 10 years. The start of the 2020 lockdown is represented by the vertical blue line.”

- **DISCUSSION (p9, l.13):** “ A prominent question raised by these findings is whether the smell and taste changes observed in our study are solely related to COVID-19 or whether they can be explained by other temporal patterns, like seasonal illnesses or allergies. To the best of our knowledge, there are no existing studies that have explored the dynamics of sudden anosmia (as in COVID-19) throughout the year in France. Relationship between olfactory disturbances and seasons have been reported in Korea, Germany or US with a moderate increase of anosmia prevalence in spring17–19. Although the cyclical pattern of smell/taste changes might overlap, the amplitude of reported changes (either due to allergy or viral affection) were very limited

compared to the present report. To further rule out the possibility, we examined whether the annual peak of allergies in France could explain the peak of smell and taste changes observed here. In analyzing existing French governmental data, we found that the annual peak of allergies in France occurred around week 30 (beginning of summer), multiple weeks after the observation window of the present study (from week 5th to week 20th, Supplementary figure 3). Further, the French national aerobiological surveillance network (RNSA), which follows pollen concentration in the atmosphere, has also indicated the first week of lockdown was very low risk for seasonal allergies²⁰. Additionally, when considering Google Trends data, we did not observe any similar peaks in queries for smell/taste loss in the corresponding time period in previous years. Finally, a comparative study in Israel by Karni et al. showed that in COVID-19 suspected patient the frequency of smell change is almost ten times higher in a COVID-19 positive patients (68%) than in COVID-19 negative (8%). Considering that most of the participants of the present study are diagnosed with COVID-19 and that their description of a sudden loss of smell/taste is consistent with the now typical presentation of COVID-19 symptoms, it is highly probable that COVID-19 infection is the main reason of their smell and taste change. Collectively, these data suggest the peak of smell and taste changes studied here are more consistent with sudden COVID-19 viral infections rather than an artifact due to seasonal illnesses.”

8. *This is not essential but would strengthen the study – If it can be demonstrated that smell/taste loss indicator can show early signal of resurgence (e.g. in Sweden).*

While we show data from France, Italy and the UK, we agree with the reviewer that more replication and validation would strengthen the study. One limitation is the need for important national homogeneous coverage to get such results. We hope that with the publication of this first step we will be able to propose an international standardised tool to replicate and validate these results as well as to assess more clearly participation bias. We agree with the reviewer, and advocate for such study in the conclusion.

- **Change in DISCUSSION - Last paragraph (p.14, l.7):** *“In summary, we propose that an increase in the incidence of sudden smell and taste change in the general population may be used as a valuable minimally invasive indicator of coronavirus spread in the population. To formally test the temporal relationship between chemosensory changes and spread of the disease, we recommend that a large-scale causal study in different countries be conducted on real-time monitoring of self-reported changes in the ability to smell or taste. Such a prospective study will allow for the creation of statistical models that can assist in prediction of future hospital admissions for COVID-19.”*

REVIEWERS' COMMENTS

Reviewer #2 (Remarks to the Author):

The authors have comprehensively evaluated and addressed my questions and need for some clarifications and revised the manuscript accordingly. Additional prospective studies are clearly needed to evaluate the generalizability of the model when it comes to its ability to in real-time monitor the epidemiology of COVID-19 in different settings and populations.

Reviewer #3 (Remarks to the Author):

The authors have addressed all comments and have further strengthened the analysis.